# Meaningful Work, Happiness at Work, and Turnover Intentions

**DOI:** 10.3390/ijerph20043565

**Published:** 2023-02-17

**Authors:** Humberto Charles-Leija, Carlos G. Castro, Mario Toledo, Rosalinda Ballesteros-Valdés

**Affiliations:** 1Instituto de Ciencias del Bienestar Integral, Tecmilenio University, Monterrey 64909, Mexico; 2Business School, Tecnológico de Monterrey, Monterrey 64849, Mexico

**Keywords:** meaningful work, happiness at work, purpose in life, job satisfaction, turnover intention

## Abstract

It has been documented that there is a positive relationship between a worker’s subjective well-being and productivity, and individuals who are happy in their work have a better attitude when performing activities: happier employees are more productive. Turnover intention, on the other hand, may arise from various factors rather than merely the need to increase a salary, as the traditional economic theory states. The fact that the work performed does not contribute to the worker’s life purpose, that there might be a bad relationship with colleagues, or else might play a role in the search for a new job. This study aims to show the relevance of meaningful work in happiness at work and turnover intention. Data from 937 professionals, in 2019, in Mexico were analyzed. Regression analyses were used to assess the impact of meaningful work on happiness at work and turnover intention. Results show that meaningful work, feeling appreciated by coworkers, and enjoyment of daily tasks significantly predict happiness at work. A logit model showed that having a job that contributes to people’s life purpose, feeling appreciated, and enjoyment of daily tasks reduces turnover intention. The main contribution of the study is to identify the importance of elements of purpose and meaning in the work context, contributing to economic theory. Limitations include the use of single items from a more extensive survey, which might diminish the validity and reliability of the constructs under scrutiny. Future directions point towards the need for more robust indicators of the variables of interest, but the findings emphasize the importance of research focused on the meaning workers attribute to their own work and the effects this attribution might have on their own wellbeing, organizational results, and productivity, including a return of investment (ROI) indicators.

## 1. Introduction

From a traditional economic standpoint, work is an exchange of time and effort for income [1]. However, it can be so much more. People are constantly facing challenges at work, putting their skills and abilities up to be tested. At work, people can find moments of satisfaction and achievement, and they can establish social relationships that contribute to their sense of well-being. Work can be a source of satisfaction, even greater than spare time. Researchers have found that if leisure time is devoted to solitary activities, low physical activity, or low intellectual effort, such as watching television, it does not generate a significant contribution to people’s wellbeing [2].

Work can be a source of personal development and recognition. When colleagues, family, and the community show appreciation, recognition, and value for people’s jobs, workers can form a more valuable perspective on their work. A job increases in value for employees when others recognize that their activity is having a favorable impact on their environment and that they are achieving something greater than themselves [3]. Work can be an element that allows individuals to reach their full potential [4], enabling them to be more productive and perceive daily activities as a source of enjoyment and not only as a sacrifice.

Bentham considered that the forces that dominated man were pleasure and pain. Historically, in economics, work has been considered a pain-generating element, or “disutility” [5]. Time at work represents an opportunity cost for people, which can be seen as a disutility. According to neoclassical economics literature, the time spent working represents hours that are not used in leisure and are not a source of joy or utility [4]. The principal-agent theory states that workers will try to avoid their tasks, so management should establish strategies to monitor them [4]. Current mainstream microeconomics models consider the hours dedicated to work as a sacrifice done to obtain income and enable to allocate it to consumption, where the utility will finally take place [6]. However, in recent years, there has been an interest in economics to see the human being beyond the vision of homo economicus [7].

This research incorporates theoretical approaches from sciences focused on the well-being of individuals, such as positive psychology and happiness economics [8]. Positive psychology has been defined as the scientific study of the strengths that enable individuals and communities to thrive. The field is founded on the belief that people want to lead meaningful and fulfilling lives, to cultivate what is best within themselves, and enhance their experiences of love, work, and play [9]. Some of their principal exponents suggest that the workplace must have room to thrive instead of suffering [10]. This study is motivated by the question: What is the effect of recognizing one’s own work as meaningful on people’s happiness at work? A second research question focuses on individuals’ turnover intention. The question is the following: What effect does meaningful work have on a personal level on the employee’s turnover intention?

In economics, meaningful work has been little studied as an element that generates utility. Hence, the present study seeks to cover this aspect. The topic is relevant because meaningful work and happiness at work are elements that contribute to the productivity and permanence of employees in the workplace. Labor turnover represents costs for organizations and national economies, and identifying the elements that increase meaning with work and happiness at work can help increase productivity levels. The results of the study are expected to provide guidelines to economists and organizational managers regarding meaningful work as a point of attention to increase happiness at work and decrease turnover intention. The research differs from the others in that it focuses the analysis on professionals by exclusively integrating graduates of the professional and master’s levels from a private university in Mexico.

Labor turnover is a severe problem for organizations because it represents costs and loss of productivity [11]. Retaining staff is one of the main challenges of human resources departments worldwide. However, measuring turnover costs is a methodological challenge that is beyond the scope of this study [12].

When a collaborator leaves the organization, the company loses the human capital that the worker had. The worker takes away technical knowledge and organizational understanding. Given this, the organization incurs expenses to find a replacement. The organization must follow a process to cover the available vacancy, and during this period, the organization stops receiving the benefits produced by the collaborator who left. Once the vacancy is filled, the new employee does not have the same organizational understanding as the previous one. The new person goes through an adaptation process where he/she does not generate the maximum profit. When a worker leaves the organization, possible future benefits from previous investments in the human capital of that collaborator are lost [12]. To cover the needs left by the vacancy, the organization may find it necessary to request efforts or overtime from the work team where a worker is missing. Causing discontent among the remaining staff. The arrival of a new worker will force the company to incur training expenses. Finally, staff turnover can reduce customer satisfaction.

People will leave expecting a higher return on their human capital. Alternatively, they could leave hoping for greater happiness at work in a different organization. On the other hand, employees who want to leave the organization are less productive. If employees are thinking of leaving the organization, they are less focused on the activities they perform; they have fewer incentives to perform optimally. Employees who know that they have little time left in the organization are less willing to establish bonds with their colleagues and obtain the greater approval of their superiors. Employees are more productive if they want to grow within the organization.

This study is based on a theoretical framework of subjective well-being; that is, it understands individuals as the best judges of their well-being [7] and focuses on the job satisfaction of 937 Mexican professionals. The study considers happiness at work as a subjective evaluation of the work environment [13]. Previous studies show that happy workers are more pragmatic, have fewer absences, are more cooperative, friendly, and more willing to help others [14].

From an experiment using the popular “Lego” game was identified that it may be more important for individuals to perform an activity with meaning and transcendence rather than monetary remuneration [15]. Accordingly, it can be affirmed that there is an intrinsic value in work [13]. It has been proposed that there are at least two important elements to identifying meaningful work: one is that it be recognized, and the other is that it has a purpose [15]. Other authors suggest that this work should contribute to a personal life purpose, as well as influence beyond the person itself [3]. The recognition does not need to be monetary; it is enough that another member of the organization (or society) expresses his appreciation for the work carried out. As far as purpose is concerned, it is said that the work is linked to a greater objective. Likewise, it has been seen that people with more routine jobs find less gratification in their work activity and that the lack of meaning in labor causes a higher reservation wage among workers. Therefore, when employees face similar jobs, people who find less meaning in a job will ask for a higher salary to carry out the activity [15]. Thus, companies that do not offer jobs with a purpose or that do not contribute to the purpose of people’s lives will find it more difficult to fill their vacancies and retain workers.

Another point to highlight in this regard is the issue of technology at work. A routine job is less rewarding, and an interesting job is more meaningful [13]. The most significant tasks are those that the collaborators will want to carry out with greater determination; if the worker bears in mind that an activity is fundamental in the productive process and represents a greater contribution to the well-being of others, he will carry it out with greater care and diligence [15].

There is no accepted definition of meaningful work [16]. Although neoclassical economic theory does not incorporate meaningful work as an element associated with the decision to work and, in general, with the level of productivity, Marx pointed out that work should have a purpose for the worker beyond material needs [13]. In Latin American culture, people can have the perception that the harder a job is, the more significant it is [17].

When people do work that is meaningful to them, they have a higher level of commitment [13]. Thus, finding meaning in work can lead individuals to dedicate themselves to their activity with more vigor, dedication, and absorption; that is, they become fully engaged in the task they are carrying out [18,19].

As with other indicators of well-being, it is up to the individual to determine whether her work is significant. Meaningful work is based on subjective criteria; however, that does not imply that it cannot be measured. People can determine whether they consider an activity worth doing or not. Building on this premise, interesting results can also suggest a dark side of meaningful work; specifically, people with meaningful work who cannot; specifically, people with meaningful work who are unable to fully employ their skills and abilities may be at particular risk for poorer well-being [20].

The benefits of meaningful work can be approached from the fact that heavy jobs involving fatigue and unpleasant activities can be seen as meaningful and transcendent jobs. Examples of this can be nursing, teaching, or cleaning in a hospital. The issue became more relevant in the year 2020 when humanity suffered a pandemic [21]. In this context, in the periods of greatest severity, many economic activities stopped, and only essential activities such as garbage collection, the police, supermarkets, and medical and hospital services were authorized [22].

A positive correlation has been identified between meaningful work and satisfaction with working conditions, health satisfaction, job involvement, and job enthusiasm, and a negative correlation with job stress [13]. Meaningful work has been measured based on two components: “if the activities of the working day give a feeling of a job well done” and if the employee “has the feeling of doing useful work” [13]. Empirically, autonomy, competence, and relationships are nearly five times more important to job meaning than compensation, benefits, career advancement, job security, and hours worked [13]. At the international level, research has been carried out mainly in psychology; this study seeks to highlight the relevance of the subject for economic science. The objective of the present study is to identify how much a meaningful job contributes to the job happiness of the workers, as well as to their turnover intention.

On the other hand, turnover intention is a response of individuals to specific conditions of the firm. Traditional economic theory would state that a worker would seek to change jobs to improve their salary. However, several studies have found that this is not the only reason someone seeks new job opportunities. The turnover intention may arise from the fact that the work performed does not contribute to the worker’s life purpose, that there is a bad relationship with co-workers, that the work is boring, or does not represent appropriate challenges for the worker. In previous studies, meaningful work has been identified as having a mediating effect between negative work conditions and turnover intention [23].

According to Frankl, meaningful work is associated with purpose and reason for living, both with vocation. Meaningful work has a negative correlation with the intention to leave the job and the organization [24]. A significant contribution of the present study is that it goes beyond correlation, and a logit model is evaluated.

Previous studies have associated turnover intention with psychological well-being [25,26]. One of the main components of psychological well-being is life purpose [27,28]. People can understand their life purpose in different ways, such as a project to be carried out, that is, a goal to be achieved. They may also see it as a reason to get up in the morning. It can be something that gives meaning and significance to their existence. In any case, the purpose of life has a subjective component that is not visible to the gaze of the researcher. However, there are strategies to approach it [29].

If the job is aligned with people’s life purpose, people will want to continue working in that organization [25]. If the work matches the workers’ life purpose, it is more likely that they can use their strengths and skills for several hours every day; this allows them to experience flow periods [30], where the challenge level is similar to the skill level.

## 2. Materials and Methods

In the item used for this study, it is possible to distinguish the factor of personal ambitions on the transcendence of the activity carried out. It is not talking about “work having a purpose beyond the person,” but that “work contributes to the purpose of the person.” Previous authors have pointed out that when a person identifies that their job has a higher purpose, they are less likely to want to quit a job.

Although it is an exploratory study, the nature of the hypotheses is relational, so we set out to determine if those important aspects of work activity in organizations, such as happiness at work and turnover intention (measured as job search), can be explained by all independent variables considered. In other words, the independent variables under analysis are presumably causal/directional determinants of happiness at work and turnover intention. Thus, rejecting null hypotheses so formulated represents evidence in favor of the systematic explanatory power of each independent variable over these two dependent variables.

### 2.1. Participants

We worked with a sample of 937 graduates of bachelor’s and master’s programs from the private institution Tecmilenio University in Mexico, who agreed to complete the usual measurement procedure that the university carries out annually for each generation. Respondents did not receive payment for answering the survey. They received the questionnaires as part of the processes of the university’s linkage with its graduates (Appendix A). Tecmilenio University was chosen because the research team works at the Instituto de Ciencias de Bienestar Integral (Institute for Happiness and Wellbeing), which is part of it; another important reason is that Tecmilenio has periodic surveys of information and allows researchers to use their data for academic purposes. Therefore, a convenience sampling method was used; all data were obtained from participants who voluntarily answered many different surveys and provided explicit consent for the use of the information. The original sample was 1208 graduates. The study focuses on the 937 graduates who currently have a job and shared information about income. Respondents’ ages varied from 22 to 65 years.

### 2.2. Procedure

The questionnaire was launched online through an electronic link to the university’s platforms and disseminated via email. The university’s web page held an invitation to participate along with a careful description of the aims of the assessment and instructions for completion.

All ethical aspects of data processing were duly complied with, as it is an internal measurement process of the university. The information about all the participants is anonymous and confidential. The research team only had access to data such as gender, age, and whether participants are currently employed or not, in addition to the answers to the questions described in the following section.

The research team only had access to the databases with all the information already filled in, and without creating the survey themselves, they didn’t have to appeal to a proper IRB approval; as a matter of fact, the University gathers and uses this information for internal control purposes only; therefore, there is no current IRB approval on this matter. However, the way information was treated complied with all ethical mandates as stipulated by the Mexican Data Protection Law that allows the processing of personal data by organizations and companies in certain situations, but always with the objective of preserving the user. Nevertheless, it is necessary that there is consent or legitimate interest in the use of the data in question; therefore, a Privacy Notice was presented to every respondent, and only those who accepted it were able to proceed with the rest of the survey. No harm was inflicted on any participant, and all the information sources remain unidentified and confidential, even for the research team. Therefore, the main limitation of the study is the use of single items from a more extensive survey dedicated to follow-up graduates in their career paths, which might diminish the validity and reliability of the constructs under scrutiny and of interest for this study.

Finally, the present study followed an inductive process: the indicators were previously defined, and only those that were of interest to the research team were selected. Researchers started from the descriptive data and then performed an inferential analysis of observations from the subsample of 937 employed graduates.

### 2.3. Instruments & Indicators

Some questions are based on the questionnaire carried out quarterly by Mexico’s National Institute of Statistics and Geography (INEGI in Spanish) as part of the Expanded Self-Reported Well-being Questionnaire (BIARE) from the year 2014 and an adaptation to the Spanish language of the well-being at work questionnaire [31,32]. In addition, some items were adapted from validated scales [33,34,35]. 

Two dependent variables were used for the investigation: happiness at work and turnover intention, measured by a single item each. The first one is a proxy for happiness at work, “During a working week, I usually feel happy” on a scale of one to seven (ranging from never to always). The second is a question about turnover intention, which corresponds to the statement “I am currently looking for a job” and is answered dichotomously with the options yes or no. Previous research has pointed out that happiness at work is a broader construct than job satisfaction [8]; therefore, the first dependent variable of the study was happiness at work, not job satisfaction.

The independent variables were purpose in life, a job that brings you closer to your purpose, a job deemed valuable, enjoying one’s job, feeling appreciated by coworkers, and happiness at work. They were all measured by asking the participants to indicate the level of agreement or disagreement, on a scale of 0 to 10, with the following phrases: “I feel that I have a purpose in life,” “For a week work, I usually enjoy my work,” “I feel appreciated by my co-workers.” Two questions focused on evaluating the level of meaning that each participant attributes to his work: “my work brings me closer to my life purpose,” “I feel that what I do in my work is valuable and useful” inspired, both in a previously validated scale [32]. Finally, the graduates were asked about their annual income (in US Dollars) within the categories defined by the University: less than $6000, $6001 to $9000, $9001 to $12,000, $12,001 to $15,000, $15,001 to $18,000, $18,001 to $24,000, $24,001 to $30,000, $30,001 or more. Sex (gender identity), age, and income group (considered only for those participants with an active job) were treated as confounding variables. Econometric analyzes were carried out using inferential statistics, as suggested by previous research [36].

### 2.4. Analytical Procedures & Statistical Analyses

The chi^2^ statistic was used to test the independence of job search and happiness at work concerning the independent variables of this study.

Once the variable’s association was tested and established, two Logit models were carried out to test the relational-directional hypothesis.

An ordered Logit model was performed to determine the explanatory level of the variables of interest (enjoying one’s job, having a life purpose, having a purpose-oriented job, feeling happy at work, feeling appreciated by colleagues, and considering one’s work as valuable) on the intention to actively search for a new job. In this first model intended to establish job search determinants, happiness at work was deemed as a factor, as will be noticed by observing the inclusion of the items related to enjoying one’s job and feeling happy at work in the results section.

On the other hand, happiness at work [13] was treated as a dependent variable in the second-ordered Logit regression model, performed to determine the explanatory level that the independent variables (purpose in life, a job that brings you closer to your purpose, a job deemed valuable, and feeling appreciated by coworkers) have on it. 

For all regression analyses on intention to turn over (active job search) and on happiness at work, overall observations were controlled for gender, age, and income level. All analyses were performed using STATA statistical software package, version 13.0.

## 3. Results

### 3.1. Descriptive Analyses

Descriptive statistics show that the sample was composed of 1208 initial observations, of whom only 1027 confirmed to have a job currently; 330 out of those, (32.13%) scored 10 on a 1–10 scale for happiness at work, 308 (29.99%) scored 9, and 389 (37.88%) scored 8 or less. These 1027 participants answered if they were actively searching for a job: 434 said yes (42.26%) and 593 (57.74%) said no; only 1001 reported their genre identity, 329 women (32.87%) and 672 men (67.13%). Age ranged from 21 to 65 years old (answered by n = 1002). Variations in the number of observations (n) were due to missing data or people refusing to answer certain questions. Details about samples are described in Table 1.

Below are three graphs that represent some variables of interest for the study. Figure 1 describes the age pyramid by sex. It is observed that the base of the pyramid is greater in the case of women. As can be seen, close to 80 of the respondents are under 40 years of age. Figure 2 shows almost half of the sample, for both men and women, are graduates of a bachelor’s degree, while the other half graduated with a master’s degree; the proportion of specialty graduates is less than 1%, both for men and for women (the survey does not collect responses from high school graduates). Figure 3 shows information about the annual income level. It is noted that 13% of the women in the sample belong to the lowest income group (less than $6000 dollars per year), while only 5% of the men report this situation. It can also be seen that 17% of men reach the highest level of income (above $30,000 dollars annually), and only 8% of women have achieved that level of income. This situation can be associated with the fact that a greater number of women surveyed are under 30 years of age and have spent less time in the labor market.

### 3.2. Inferential Analyses

Association between variables was assessed firstly to determine factor variables (determinants) to be used in directional, deterministic statistical models of happiness at work and turnover intention (logistic and ordered logistic regressions). The number of observations varied from the independence/association analyses to logistic and ordered logistic regressions (performed each time with an n = 937) due to missing data and participants’ refusal to answer some questions. 

### 3.3. Happiness at Work

Independence between variables was assessed through chi^2^ tests yielding an association between *happiness at work* and enjoying one’s job (chi^2^ = 2800, *p* = 0.000; n = 937) and having a purpose in life (chi^2^ = 698.31, *p* = 0.000; n = 937), and having a job that brings you closer to your purpose in life (or life purpose oriented job: chi^2^ = 900.18, *p* = 0.000; n = 937) and having a job deemed as valuable (chi^2^ = 1100, *p* = 0.000; n = 937).

*Confounding/control variables*. None of the control variables (age, gender, and income) had a significant association with happiness at work, nor did they interact with the other variables in predicting or explaining it.

### 3.4. Odd Ratios of Having Happiness at Work

Regarding factor and outcome variables, results show that controlling for age, gender, and income, for a one-unit increase in enjoying one’s job, the odds of having happiness at work increase by a factor of 6.06 (OR = 6.06, z = 19.33, *p* = 0.000 [95% CI: 5.05, 7.28] n = 937). For a one-unit increase in feeling appreciated by coworkers, the odds of having happiness at work increase by a factor of 1.27 (OR = 1.27, z = 4.29, *p* = 0.000 [95% CI: 1.13, 1.41] n = 937). For a one-unit increase in life purpose, the odds of having happiness at work increase by a factor of 1.22 (OR = 1.22, z = 3.67, *p* = 0.000 [95% CI:1.09 1.36] n = 937). For a one-unit increase in deeming one’s job as valuable, the odds of having happiness at work increase by a factor of 1.15 (OR= 1.15, z = 2.25, *p* ≤ 0.05 [95% CI: 1.01, 1.30] n = 937).

The likelihood ratio chi-square (chi^2^ = 1193.84, *p* = 0.000, n = 937) tells us that our model fits significantly better than an empty model (i.e., a model with no predictors); the explained variance for the overall model was r^2^ = 0.39.

### 3.5. Marginal Effects on Happiness at Work

The predicted probabilities of different outcomes for happiness at work are presented as follows:

#### 3.5.1. Margins at the 10th Outcome

Marginal effects analyses show that the whole model with margins at the tenth outcome and all factors included predicts the probability of having happiness at work at 10.7%. On the other hand, enjoying one’s job increases the probability of being happy at work by 17% (0.17, z = 12.58, *p* = 0.000); feeling appreciated by coworkers increases the probability of being happy at work by 2.3% (0.023, z = 4.18, *p* = 0.000); having a purpose in life increases the probability of being happy at work by 1.9% (0.019, z = 3.66, *p* = 0.000), and having a job deemed as valuable increases the probability of being happy at work by 1.3% (0.013, z = 2.24, *p* ≤ 0.05). See Table 2.

#### 3.5.2. Margins at the 9th Outcome

Marginal effects analyses show that the whole model with margins at the ninth outcome and all factors included predicts the probability of having happiness at work at 53%. At the same time, enjoying one’s job increases the probability of being happy at work by 24% (0.24, z = 8.59, *p* = 0.000); feeling appreciated by coworkers increases it by 3% (0.03; z = 3.97, *p* = 0.000); having a purpose in life increases it by 2.7% (0.027, z = 3.59, *p* = 0.000), and having a job deemed as valuable increases it by 1.8% (0.018, z = 2.19, *p* ≤ 0.05). See Table 2.

#### 3.5.3. Margins at the 8th Outcome

Marginal effects analyses show that the whole model with margins at the eighth outcome and all factors included predicts the probability of having happiness at work at 31%. Interestingly, starting from here on (margins below the ninth outcome), predictions begin to turn into opposites by changing signs into negative; thus, enjoying one’s job decreases the probability of being happy at work by 33% (−0.33, z = −13.06, *p* = 0.000); feeling appreciated by coworkers decreases it by 4% (−0.04, z = −4.22, *p* = 0.000); having a purpose in life decreases it by 3% (−0.03, z = −3.74, *p* = 0.000), and having a job deemed as valuable decreases it by 2% (−0.02, z = −2.24, *p* ≤ 0.05). See Table 2.

#### 3.5.4. Margins below the 8th Outcome

On the other hand, marginal effects analyses show that the whole model with margins below the seventh outcome or less predicts happiness at work in small amounts and are either not statistically significant in most cases or direct (positive-signed) in determining happiness at work.

### 3.6. Turnover Intention

Independence between variables was assessed through chi^2^ tests yielding an association between the intention to turnover (*active job search*) and enjoyment of work (chi^2^ = 150.96, *p* = 0.000; n = 937), and life purpose (chi^2^ = 20.01, *p* ≤ 0.05; n = 937), and life purpose-oriented work (chi^2^ = 136.80, *p* = 0.000; n = 937), and happiness at work (chi^2^ = 105.99, *p* = 0.000; n = 937) and having a job deemed as valuable (chi^2^ = 69.94, *p* = 0.000; n = 937), respectively.

#### 3.6.1. Confounding Variables

Age (n = 937) and gender (n = 937) did not have a significant association with turnover intention (*active job search*). The income level, on the other hand, was associated with turnover intention (chi^2^ = 41.65, *p* = 0.000, n = 937).

#### 3.6.2. Odd Ratios of Turnover Intention

Results show that even though age and gender were not associated with the outcome variable when controlling for age, gender, and income, gender had a significant interaction with the explanatory variables of the intention to turnover within the Logit regression model (OR = 0.70, z = −2.08, *p* ≤ 0.05 [95% CI: 0.50, 0.97] n = 937); in other words, when being a woman, the odds of being actively searching for a new job (versus not doing so) multiply by a factor of 0.70. The income level, on the other hand, also had a significant interaction with the explanatory variables of the intention to turnover only from a salary of $24,000 USD per year onwards (OR = 0.305, z = −2.95, *p* = 0.003 [95% CI: 0.13, 0.67] n = 937); in other words, starting from that amount of income per year, the odds of being actively searching for a new job (versus not doing so) multiply by a factor of 0.305.

Regarding factor and outcome variables, for a one-unit increase in having a purpose in life, the odds of being actively searching for a new job increase by a factor of 1.36 (OR = 1.36, z = 4.05, *p* = 0.000 [95% CI: 1.17, 1.58] n = 937) 

Conversely, for each increase in one unit in feeling appreciated by coworkers, the odds of being actively searching for a new job multiplied by a factor of 0.82 (OR = 0.82, z = −2.88, *p* ≤ 0.01 [95% CI: 0.71, 0.93] n = 937). For a one unit increase in having a job that brings you closer to your purpose in life, the odds of being actively searching for a new job multiplied by a factor of 0.74 (OR = 0.74, z = −4.59, *p* = 0.000 [95% CI: 0.65, 0.84] n = 937). For a one unit increase in enjoyment at work, the odds of being actively searching for a new job (versus not doing so) multiply by a factor of 0.632 (OR = 0.63, z = −4.23, *p* = 0.000 [95% CI: 0.51, 0.78], n = 937).

The explained variance for the overall model was r^2^ = 0.16, and the likelihood ratio chi-square (chi^2^ = 211.76, *p* = 0.000, n = 937) tells us that our model as a whole fits significantly better than an empty model (i.e., a model with no predictors).

### 3.7. Marginal Effects for Turnover Intention

The predicted probabilities at means for turnover intention are presented as follows: 

Marginal effects analysis shows that the whole model with margins at means and all factors included predicts the probability of actively searching for a new job at 40% (n = 937). At the same time, being a woman decreases it by 8% (−0.08, z = −2.07, *p* ≤ 0.05), earning an income higher than $24,000 USD per year decreases it by 3% (−0.03, z = −4.06, *p* = 0.000), enjoying one’s job decreases it by 11% (−0.11, z = −4.35, *p* = 0.000), having a job that brings you closer to your purpose in life decreases it by 7% (−0.074, z = −4.71, *p* = 0.000) and feeling appreciated by coworkers decreases it by 4.5%. (−0.045, z = −2.77, *p* = 0.006) Conversely, having a purpose in life increases it by 0.7% (0.07, z = 4.24, *p* = 0.000); see Table 2.

## 4. Discussion

Referring to happiness at work, results indicate that gender was not statistically significant in predicting it, nor were income level and age. Consistent with the Positive Psychology theory, having a life purpose explains happiness at work in a significant and positive way. The belief in having useful and valuable work shows a positive and significant effect on happiness at work. The variable that best explains happiness at work is the enjoyment of the activities performed; that is, the more the individual enjoys daily tasks, the greater the level of happiness at work. The second element that represents a greater weight in happiness at work is co-workers; when they help the worker feel appreciated, the chances of someone being happy at work increase. The model explains the variance moderately and significantly; 39 out of each 100 employees report happiness at work based on all the independent variables. Surprisingly, having a job that brings the individual closer to his or her life purpose does not have a significant effect on happiness at work.

The second model seeks to explain turnover intention. Results indicate that women in this sample have less turnover intention and are probably more stable in terms of job permanence, whilst an income above $24,000 USD per year diminishes the intention to seek another job; age, on the other hand, does not exert a significant influence on the search for another job.

Interestingly, people who have a high purpose in life have more intention to leave their job. However, if a worker considers that his/her job does not contribute to achieving their life purpose, their intentions to look for another job increase. A surprising result was that the item related to useful and valuable work was not significant in predicting turnover intention. All this can be explained by the economic theory that proposes the worker as a selfish agent, more concerned with the purpose of his personal life than with the impact of his work on others. Consequently, it is observed that employees consider leaving a job that does not contribute to their life purpose; however, the fact that the job is not seen as useful and valuable does not trigger the intention to take specific actions to leave their job. Similarly, with the regression model regarding happiness at work, the variable that best explains turnover intention is the enjoyment of one’s job. Enjoying one’s job significantly reduces the chances that a person will try to leave their job.

Once again, the relational factor is crucial to explain the intention to leave a company. Results also show that feeling appreciated significantly reduces an employee’s turnover intention.

It is important for people to understand the impact their work can have on their life purpose. A relevant contribution of the present study is that working in a job that does not contribute to achieving a person’s life purpose will increase their desire to change jobs. A fundamental element to highlight in this study is the one referring to the enjoyment of daily activities. It is convenient to assume that more interesting jobs will generate happier employees.

In consequence, organizations must try to ensure that workers find enjoyment in the activities they perform; one tactic may be the use of job crafting techniques, strategies for workers to adapt their jobs for their own benefit [37,38].

Many differences in productivity between employees start with subjective well-being, such as personal relationships and happiness at work [14], or the meaning employees attribute to their own work. In the present study, the relevance of meaningful work, happiness at work, and turnover intention were also explored.

If workers find their job interesting, they have more intrinsic motivations, carry out activities with greater determination, and perform activities that go beyond their contractual obligations, a concept that the organizational literature has called “extra-role performance” [39], often referred to as “the extra mile” that some employees are willing to offer when they are determined and engaged with their jobs.

This study identified that if employees work in a job that does not contribute to their life purpose, they will be more likely to seek another job. This finding has various practical, theoretical, and methodological implications.

On a practical level, this analysis contributes to the understanding of the non-pecuniary motivations that lead people to seek work in other organizations. Although economic theory highlights that salary is a central component of the employment decision, this research brings to the table a subject that has been little explored in economic science: meaningful work. Based on what was identified in this study, it can be pointed out that companies need to research the contribution that organizational objectives have to the life goals of their workers.

People will be more motivated to work in a firm if the activities they carry out allow them to get closer to the long-term goals they have in their lives. Measuring people’s purpose in life is a methodological challenge. In this study, such a variable is measured using a single item. Future studies can further explore whether such an item is useful for predicting turnover intention in different contexts.

A valuable contribution of the study is that it measures two variables of interest in reference to meaningful work: its contribution to the purpose of personal life and its value and usefulness to others. In this sense, the research shows two interesting results. The first, expected, is that a reason to change jobs is to carry out an activity that does not contribute to the worker’s purpose in life. The second, surprisingly, is that a job of little value to others does not motivate the worker to look for another job, perhaps due to the weight some other variables have for the worker. The latter is interesting because it contradicts the postulates of positive psychology; however, it does not mean that the value that one assigns to their work is not important, but rather that there are other variables that are more relevant; More evidence needs to be gathered around this point.

The item referring to happiness at work used in this study generates better results than the items that measure job satisfaction in previous studies [40]. This item, along with the independent variables (related to positive psychology), shows a significant increase in explained variance. In consequence, the new item can be a good contribution to measuring various elements related to subjective well-being in the work context.

One limitation of this study is that it focused only on graduates of a private university in Mexico. It would be worthwhile to extend the effort to larger populations so that data can better represent a full regional or national perspective.

Other limitations include the use of single items from a more extensive survey which might diminish the validity and reliability of the constructs under scrutiny. We recommend being cautious with these findings since we used single indicators rather than a fully validated and standardized instrument (variables were not measured as multicomponent constructs), yet all indicators were inspired by either solid theoretical postulates or came from validated sources such as the BIARE survey from Mexico’s National Institute of Statistics and Geography. There is value in aggregating the items within a larger construct for future studies. 

## 5. Conclusions

Previous studies have started from the hypothesis: “Having a life purpose reduces turnover intentions.” The present study goes further: the authors proposed that it is not enough for a person to have a life purpose of wanting to stay in a job. People need to feel that their job contributes to their life purpose. That is why it is essential that organizations properly communicate to employees how working in a certain position will allow them to achieve their mission in life. Therefore, companies must identify the strengths, skills, and goals of workers so that they can offer solutions aimed at those goals.

It was also found that feeling appreciated by co-workers and enjoying the activities carried out also make a significant contribution to happiness at work, remarking the importance of subjective variables in determining it.

This research was motivated by the questions: What is the effect of recognizing one’s own work as significant on people’s happiness at work? To what effect does the significance of personal work have on turnover intention? Results allowed us to appreciate that meaningful work generates positive and significant contributions to the happiness at work of the respondents and reduces turnover intention.

This supports the series of recent efforts to incorporate elements of meaningful work into economic theory [13,15]. Hours spent working do not necessarily are disutility. Utility and well-being can be obtained from work if it follows a purpose, and it is relevant for life’s meaning, if it is lived in an environment of respect, partnership, and appreciation, and if the tasks allow individuals to match their skills and abilities with the job challenges and demands.

A relevant finding of the study is to show that meaningful work can be independent of income. From the economics tradition, the main incentive for a worker is the pecuniary element for both work and the rest of the economic decisions. In this research, the pecuniary component was irrelevant in its influence on happiness at work and only significant when earning above $24,000 USD yearly regarding turnover intentions. Data showed that elements such as life purpose, job relationships, and meaningful work are aspects that are more relevant for individuals than wages. The study is consistent with previous work that has highlighted the aspect of psychological needs as a key determinant of happiness at work [8]. This is very good news for employers who can integrate practices to achieve greater happiness at work without affecting their cost structure (salaries or other benefits for employees).

## Figures and Tables

**Figure 1 ijerph-20-03565-f001:**
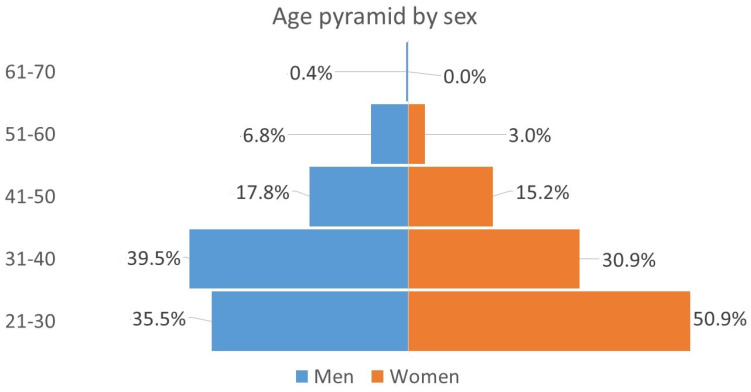
Age pyramid by sex.

**Figure 2 ijerph-20-03565-f002:**
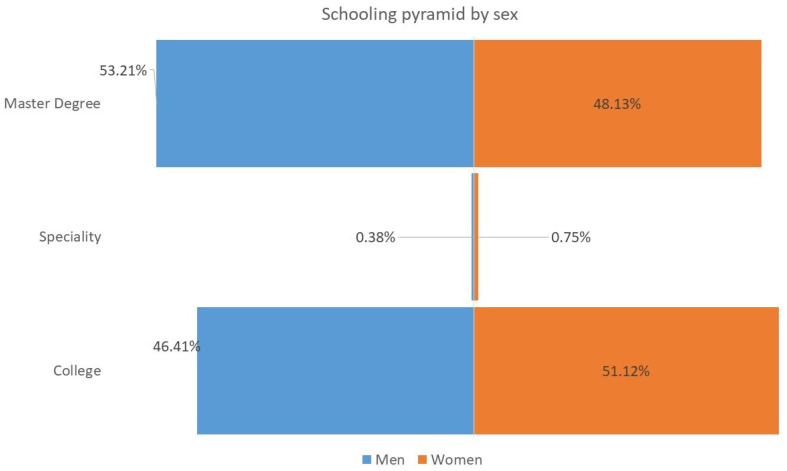
Schooling pyramid, by sex.

**Figure 3 ijerph-20-03565-f003:**
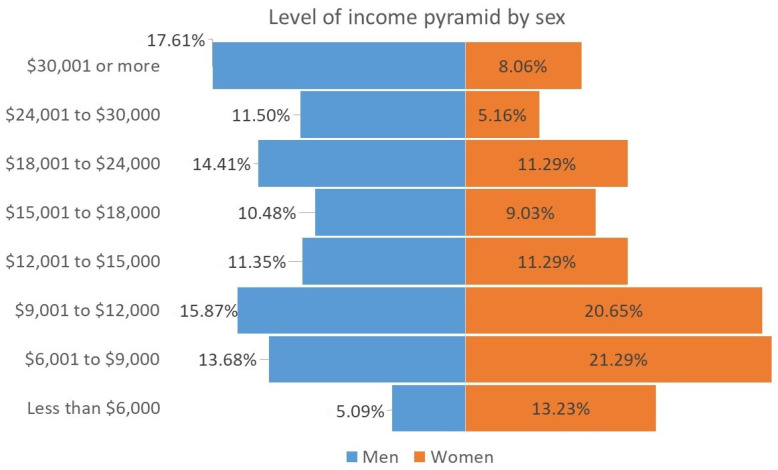
Yearly income level pyramid by sex.

**Table 1 ijerph-20-03565-t001:** Sample information.

Valid responses	1208
Employees	1027
Employees who shared information about gender	1001
Employees who shared information about gender and income	937

**Table 2 ijerph-20-03565-t002:** Marginal analyses: Predicted probabilities of Turnover Intention and Happiness at Work at factor outcomes 8, 9, and 10.

	Job Search Factors: Predicted Probabilities at Means	95% CI	Happiness at Work Factors: Predicted Probabilities at Outcome 10	95% CI	Happiness at Work Factors: Predicted Probabilities at Outcome 9	95% CI	Happiness at Work Factors: Predicted Probabilities at Outcome 8	95% CI
Overall margins	0.40		0.10		0.53		0.31	
Purpose in life	**0.07 *****	[0.041, 0.113]	**0.01 *****	[0.009, 0.030]	**0.027 *****	[0.012, 0.043]	**−0.03 *****	[−0.059, −0.018]
A job that brings you closer to your purpose	**−0.07 *****	[−0.105, −0.043]	0.005	[−0.004, 0.016]	0.008	[−0.005, 0.022]	−0.011	[−0.031, 0.008]
A job deemed as valuable	0.01	[−0.019, 0.058]	**0.013 ****	[0.001, 0.025]	**0.018 ****	[0.002, 0.035]	**−0.02 ****	[−0.049, −0.003]
Enjoying one’s job	**−0.11 *****	[−0.163, −0.061]	**0.17 *****	[0.145, 0.198]	**0.24 *****	[0.185, 0.295]	**−0.33 *****	[−0.386, −0.285]
Feeling appreciated by coworkers	**−0.04 ****	[−0.078, −0.013]	**0.02 *****	[0.012, 0.034]	**0.03 *****	[0.016, 0.048]	**−0.04 *****	[−0.066, −0.024]
Happiness at work	0.01	[−0.031, 0.065]	-	-	-	-	-	-
Sex	**−0.08 ****	[−0.157, −0.004]	0.006	[−0.020, 0.034]	0.009	[−0.027, 0.046]	−0.012	[−0.065, 0.039]
Age	0.00	[−0.003, 0.004]	0.000	[−0.002, 0.0009]	−0.0008	[−0.003, 0.001]	0.001	[−0.001, 0.004]
Income group	**−0.03 *****	[−0.053, −0.018]	0.003	[−0.009, 0.002]	−0.004	[−0.012, 0.004]	0.006	−0.005, 0.017

Significant values **in bold:** ** *p* ≤ 0.05; *** *p* ≤ 0.01; n = 937.

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
