# Peer review of "Meaningful Work, Happiness at Work, and Turnover Intentions"

_ijerph, 2023, doi:10.3390/ijerph20043565_

Round 1

Reviewer 1 Report

Overall, this is generally good paper – it does however present opportunities for improvement as noted – specifically, there is a need to add conflicting and divergent perspectives evident in the literature to the article (2-3 paragraphs) to ensure comprehensiveness. As well, there is a need for the author(s) to incorporate limitations to the paper as noted above within the Abstract and then at the end of the Methodology Section), as opposed to the back end of the paper. Finally, I would recommend the author(s) provide further clarity related to why a specific private university was selected; and if participants or the university received an incentive to participate in the study. By incorporating the spirit and intent of these suggested revisions, I believe the author(s) can take their paper to the next level – that is publication.

Overall an interesting paper, which presents, and discusses the relevance of meaningful work in happiness at work and the turnover intention using a theoretical lens of subjective well-being based on job satisfaction of Mexican professionals. From this perspective, the paper will garner interest from academia. From an academic standpoint the mix of current (less than 5 year peer reviewed research) presented and discussed in the paper is overall representative of the current literature. However, as an opportunity to demonstrate inclusiveness, there is need for a brief (2-3 paragraphs) on divergent and conflicting perspectives within the literature to avoid the narrow focus perception which many academic would consider as selective literature to support research agenda. Avoiding this shortcoming and to garner academic community interest, it is necessary to extend the research to add the conflicting and divergent perspectives from the literature, which in turn will ensure the paper’s overall comprehensiveness and currency to the intended audience.

From a methodology perspective, the paper is generally well designed and appropriate – the exception the requirement for further clarity related to why a specific private university was selected; and if participants or the university received an incentive to participate in the study. As well, there is a need to briefly note the limitation in the Abstract; and this may very well reflect my own personal preferences. However, in defense of this posture, the identification of limitations at the front (Abstract and at the end of the Methodology Section) enables the reader audience to understand the limitations and any caveats up front, and prior to reading the paper through and then findings limitations at the back end of the paper. It is imperative that reader(s) have the opportunity to place the paper into a relevant context in relation to what is stated / proposed by the author(s).

The results are generally presented in a clear and concise manner; there is evidence of analysis evident; the inclusion of the Table is an effective visual. As well, the author(s) have developed an acceptable linkage between the results and conclusions noted. The points noted in paper are tied together into a final coherent picture. It is evident that the author(s) have an excellent understanding of the subject area.

This is a solid paper in many respects, since it provides several opportunities for continued research in the subject area with the possibility of different streams within the research area, while providing further avenues of research potential. With respect to the practical application of the research, it presents an opportunity to enhance the depth, breadth and understanding of factors related to meaningful work and why and how meaningful work should contribute to personal economic well-being and the collective well-being of individuals.

The writing quality of the paper demonstrates scholarly flow and readability – no professional edit is required.

Confidential Comments to the Editor-in-Chief

Overall, this is generally good paper – it does however present opportunities for improvement as noted – specifically, there is a need to add conflicting and divergent perspectives evident in the literature to the article (2-3 paragraphs) to ensure comprehensiveness. As well, there is a need for the author(s) to incorporate limitations to the paper as noted above within the Abstract and then at the end of the Methodology Section), as opposed to the back end of the paper. Finally, I would recommend the author(s) provide further clarity related to why a specific private university was selected; and if participants or the university received an incentive to participate in the study. By incorporating the spirit and intent of these suggested revisions, I believe the author(s) can take their paper to the next level – that is publication.

Minor Revisions Required

Comments to the Author

A generally good paper - For consideration, I recommend the inclusion of research which briefly presents / discusses the conflicting and divergent perspectives in the literature (2-3 paragraphs), updated the Abstract to briefly note limitations, and then include the limitations following your Methodology section as opposed to the back end of the paper. Finally, I would recommend the author(s) provide further clarity related to why a specific private university was selected; and if participants or the university received an incentive to participate in the study.

If you take the time and effort to incorporate the spirit and intent of the suggested revisions to your paper, you will take your paper to the next step – that is publication and a following for your research.

Author Response

Reviewer 1

Comments (C) and Responses (R)

C

Overall an interesting paper, which presents, and discusses the relevance of meaningful work in happiness at work and the turnover intention using a theoretical lens of subjective well-being based on job satisfaction of Mexican professionals. From this perspective, the paper will garner interest from academia. From an academic standpoint the mix of current (less than 5 year peer reviewed research) presented and discussed in the paper is overall representative of the current literature. However, as an opportunity to demonstrate inclusiveness, there is need for a brief (2-3 paragraphs) on divergent and conflicting perspectives within the literature to avoid the narrow focus perception which many academic would consider as selective literature to support research agenda. Avoiding this shortcoming and to garner academic community interest, it is necessary to extend the research to add the conflicting and divergent perspectives from the literature, which in turn will ensure the paper’s overall comprehensiveness and currency to the intended audience.

R

New literature was added to the manuscript, incorporating complementary ideas to those initially exposed.

C

From a methodology perspective, the paper is generally well designed and appropriate – the exception the requirement for further clarity related to why a specific private university was selected; and if participants or the university received an incentive to participate in the study.

R

"Respondents did not receive a payment for answering the survey. They receive the questionnaires as part of the processes of linking the university with its graduates. Universidad Tecmilenio was chosen because it is an institution that has periodic surveys of information and allows researchers to use their data for academic purposes."

C

As well, there is a need to briefly note the limitation in the Abstract; and this may very well reflect my own personal preferences. However, in defense of this posture, the identification of limitations at the front (Abstract and at the end of the Methodology Section) enables the reader audience to understand the limitations and any caveats up front, and prior to reading the paper through and then findings limitations at the back end of the paper.

R

Limitations were included in the abstract and in the method.

C

It is imperative that reader(s) have the opportunity to place the paper into a relevant context in relation to what is stated / proposed by the author(s).

R

Descriptions that highlight the relevance of the paper in the current context were included.

C

The results are generally presented in a clear and concise manner; there is evidence of analysis evident; the inclusion of the Table is an effective visual. As well, the author(s) have developed an acceptable linkage between the results and conclusions noted. The points noted in paper are tied together into a final coherent picture. It is evident that the author(s) have an excellent understanding of the subject area.

This is a solid paper in many respects, since it provides several opportunities for continued research in the subject area with the possibility of different streams within the research area, while providing further avenues of research potential. With respect to the practical application of the research, it presents an opportunity to enhance the depth, breadth and understanding of factors related to meaningful work and why and how meaningful work should contribute to personal economic well-being and the collective well-being of individuals.

The writing quality of the paper demonstrates scholarly flow and readability – no professional edit is required.

R

Thanks for all your comments

Reviewer 2 Report

I would like to start by congratulating the authors for such valuable work. However, the manuscript has some shortcomings that should be corrected.

In the first place, it is necessary to cite and reference the authors of positive psychology referred to in the introduction and discussion, as well as to explain what said theory consists of and its link with this study.

Secondly, a more detailed description of the sample must be made. That is, include attributive variables such as gender, age, academic training, etc. that allow to describe the sample conveniently. In addition, it is highly recommended to describe the sample graphically, using population pyramids, graphs based on the salary received, etc.

It should be noted that the lack of graphs can be applied to the results as well. It is convenient to include graphs relating the different variables studied that visually show the results obtained with the regression lines. It is not recommended to include all the results in the text, since it is difficult to understand.

Finally, I think that the description of the variables should be improved. The study variables are not the same as the test items, but rather the items respond to the questions that are posed based on these variables. A section within the method in which the variables are specifically described can be beneficial for the clarity and understanding of the article.

Author Response

Reviewer 2

Comments (C) and Responses (R)

C

I would like to start by congratulating the authors for such valuable work. However, the manuscript has some shortcomings that should be corrected. 

R

Thanks for all your comments

C

In the first place, it is necessary to cite and reference the authors of positive psychology referred to in the introduction and discussion, as well as to explain what said theory consists of and its link with this study. 

R

Several papers regarding positive psychology and happiness economics were included.

C

Secondly, a more detailed description of the sample must be made. That is, include attributive variables such as gender, age, academic training, etc. that allow to describe the sample conveniently. In addition, it is highly recommended to describe the sample graphically, using population pyramids, graphs based on the salary received, etc. 

It should be noted that the lack of graphs can be applied to the results as well. It is convenient to include graphs relating the different variables studied that visually show the results obtained with the regression lines. It is not recommended to include all the results in the text, since it is difficult to understand. 

R

Several graphs were included.

C

Finally, I think that the description of the variables should be improved. The study variables are not the same as the test items, but rather the items respond to the questions that are posed based on these variables. A section within the method in which the variables are specifically described can be beneficial for the clarity and understanding of the article.

R

A more appropriate description of the variables was included, avoiding expressions in Spanish.

Reviewer 3 Report

Thank you for the opportunity to review the article entitled “Meaningful Work, Happiness at Work, and Turnover Intentions”.

The topic is interesting for a broad readership and the study shows relevant results. However, several concerns preclude publication in its present form.

- The background requires reorganization and avoidance of repetitions. Part of it is provided in the methods section instead of the Introduction, while the introduction is too long and repetitive.

- Accurate information of the participants is lacking; it is required to know the age distribution, the proportion of males and females, marital status, degree of education, profession and occupation.

- The description of the number of participants is both confusing and inconsistent throughout the different sections of the manuscript, including the abstract.   

- No information is provided on the validity and reliability of the instrument.

- The description of the statistical analysis is incomplete. As a result, the description of the results is confusing.

- The results could be clarified by guiding the reader through the analysis (that should be explained in the appropriate section).

- The table of results needs both editing and appropriate heading, in order to be of any use to the reader.

- The discussion is confusing and just partially contrast or explain the specific results of the study according to the international literature.   

- The conclusions are mixed-up with the discussion and limitations of the study. This section should include just the final inferences supported by the results of the study.

- It should be clarified if the participants gave consent to the use of their responses for the research study, since they provided the information while participating in an internal process of the university.

- Some sentences are written on the opposite way as intended (positive instead of negative) bringing confusion to the reader (e.g. “   If the work that a person does is aligned with their life mission, that person will not  want to continue working in that organization; otherwise, they will look for a new job that 186 can better match their meaning of life” lines 185-187).

- Review of the text is highly recommended, since authors did not remove words in a different language than English (e.g.increase in enjoyment at work (disfrutot)”& “in feeling appreciated by coworkers (apreciado)”).

- Since the authors have assessed income by Mexican pesos, an estimated conversion should be provided in a more international currency ( either U.S. dollars or Euros).

Thank you for the opportunity to Review the article entitled “Meaningful Work, Happiness at Work, and Turnover Intentions”.

Author Response

Dear reviewer, we are very grateful for the feedback you gave us. His comments allowed us to improve the work. We make adjustments to the manuscript based on your observations in conjunction with the comments of the other reviewers. The changes are in yellow.

Thanks again

Comments (C) and Responses (R)

C

The background requires reorganization and avoidance of repetitions. Part of it is provided in the methods section instead of the Introduction, while the introduction is too long and repetitive. 

R

Changed the intro to be less repetitive. Some aspects were expanded at the request of various reviewers.

C

Accurate information of the participants is lacking; it is required to know the age distribution, the proportion of males and females, marital status, degree of education, profession and occupation.

R

Several graphs were included.

C

The description of the number of participants is both confusing and inconsistent throughout the different sections of the manuscript, including the abstract.   

R

The number of observations varied from the independence/association analyses to logistic and ordered logistic regressions (performed each time with an n = 937) due to missing data and participant’s refusal to answer some questions.

C

No information is provided on the validity and reliability of the instrument.

R

In the future we suggest using validated instruments. An important limitation of the study is the fact that we did not measure variables as multicomponent constructs, therefore validity and reliability are still at stake; There is value in aggregating the items within a larger construct, for future studies.

C

The description of the statistical analysis is incomplete. As a result, the description of the results is confusing.

R

A more detailed and coherent description has been carried out

C

The results could be clarified by guiding the reader through the analysis (that should be explained in the appropriate section).

R        

Results have been put ahead the new statistical analysis description, trying to guide readers through it in a clearer fashion

C

The table of results needs both editing and appropriate heading, in order to be of any use to the reader.

R

Headings have been corrected and edited; We are still presenting confidence interval (CI) columns besides the predicted probability columns, so the reader can verify the correspondence between statistical significance and the respective Cis

C

The discussion is confusing and just partially contrast or explain the specific results of the study according to the international literature.  

R

Discussion has been modified to be more comprehensive

C

The conclusions are mixed-up with the discussion and limitations of the study. This section should include just the final inferences supported by the results of the study.

R

Conclusion has been modified to include only final inferences.

C

It should be clarified if the participants gave consent to the use of their responses for the research study, since they provided the information while participating in an internal process of the university.

R

It has been clarified and all information has been treated in compliance with all ethical mandates as stipulated by the Mexican Data Protection Law, as described in the new paragrpahs included in the document.

C

Some sentences are written on the opposite way as intended (positive instead of negative) bringing confusion to the reader (e.g. “   If the work that a person does is aligned with their life mission, that person will not  want to continue working in that organization; otherwise, they will look for a new job that 186 can better match their meaning of life” lines 185-187).

R

The sentece was clarified

C

Review of the text is highly recommended, since authors did not remove words in a different language than English (e.g. “increase in enjoyment at work (disfrutot)”& “in feeling appreciated by coworkers (apreciado)”).

R

All variables are in English

C

Since the authors have assessed income by Mexican pesos, an estimated conversion should be provided in a more international currency ( either U.S. dollars or Euros).

R

Information about income were included as dollars per year

C

Thank you for the opportunity to Review the article entitled “Meaningful Work, Happiness at Work, and Turnover Intentions”.

R

Thanks for all your comments

Reviewer 4 Report

1) The abstract should contain the problem statement before the aim of study is presented. The abstract should also mention a brief statement of the directions for future work.

2) The authors should state the sampling method used (for example, random sampling, systematic sampling, convenience sampling etc).

3) The authors should demonstrate the reliability of the instrument (for instance, Cronbach's alpha analysis). There should be factor analysis as well.

4) The hypotheses should be proposed explicitly, even if they are relational. The proposed hypothetical framework should be drawn out as well.

5) Since regression is involved, how was the normality of the standardised residuals of the dependent variable like? Was there equality of variance as well? Please satisfy these conditions before conducting a regression.

Thank you.

Author Response

Dear reviewer, we are very grateful for the feedback you gave us. His comments allowed us to improve the work. We make adjustments to the manuscript based on your observations in conjunction with the comments of the other reviewers. The changes are in yellow.

Thanks again

1)    The abstract should contain the problem statement before the aim of study is presented. The abstract should also mention a brief statement of the directions for future work. 

Problem and future directions were addressed in abstract

2)    The authors should state the sampling method used (for example, random sampling, systematic sampling, convenience sampling etc). 

The sampling method used has been stated.

3)    The authors should demonstrate the reliability of the instrument (for instance, Cronbach's alpha analysis). There should be factor analysis as well. 

Description about reliability was included in the manuscript.

4)    The hypotheses should be proposed explicitly, even if they are relational. The proposed hypothetical framework should be drawn out as well. 

The hypotheses have been explicitly proposed.

5)    Since regression is involved, how was the normality of the standardised residuals of the dependent variable like? Was there equality of variance as well? Please satisfy these conditions before conducting a regression. 

Dear reviewer: we've been consulting Cameron & Trivedi's book (2009), as well as Dunn & Smyth (2018) and we haven't been able to find a priori tests to satisfy conditions prior to conducting logit regression procedures; would you be kind enough to guide us trough this matter?

Cameron, A. C., & Trivedi, P. K. (2005). Microeconometrics: methods and applications. Cambridge university press.

Dunn, P. K., & Smyth, G. K. (2018). Generalized linear models with examples in R (Vol. 53). New York: Springer.

Thank you. 

Round 2

Reviewer 3 Report

Thank you for the opportunity to review the revised version of the manuscript entitled “Meaningful Work, Happiness at Work, and Turnover Intentions”.

The topic is interesting and the manuscript has greatly improved. Yet, the description of the number of participants is still confusing. It is highly recommended to use a flowchart and to clarify the number of subjects with complete data included in the analysis, with the description of those who did not complete the survey, in order to compare them (this information can be provided as supplementary material).    Apparently among 1208 candidates, 1027 were selected according to the main selection criteria; however, only 937 completed the survey. However, analysis was performed including subjects who provide incomplete information (risk of bias).

- Since there is no validity information on the instrument, more detail is required about its construction and standardization. Otherwise any interpretation or conclusion is weakened.

- The conclusions have not been separated from the limitations (which could be described at the end of the Discussion section).  This section should include just the final inferences supported by the results of the study.

- A second review of the language use  and the journal format is required

Author Response

Dear reviewer, we are very grateful for the feedback you gave us. His comments allowed us to improve the work. We adjust the manuscript based on your observations in conjunction with the comments of the other reviewers. The changes are in yellow.

Comments (C) and Responses (R)

C

The topic is interesting and the manuscript has greatly improved. Yet, the description of the number of participants is still confusing. It is highly recommended to use a flowchart and to clarify the number of subjects with complete data included in the analysis, with the description of those who did not complete the survey, in order to compare them (this information can be provided as supplementary material).    Apparently among 1208 candidates, 1027 were selected according to the main selection criteria; however, only 937 completed the survey. However, analysis was performed including subjects who provide incomplete information (risk of bias).

R

Table 1 presents a brief and detailed description of the sample. Some addditional modifications was made in the manuscript.

C

- Since there is no validity information on the instrument, more detail is required about its construction and standardization. Otherwise any interpretation or conclusion is weakened.

R

The origin of each item was placed in the table 3

C

The conclusions have not been separated from the limitations (which could be described at the end of the Discussion section).  This section should include just the final inferences supported by the results of the study

R

Conclusion was separated from the limitations. Conclusions only include final inferences suppoted by the results of the study.

C

- A second review of the language use  and the journal format is required

R

Authors will use the MDPI services for review of language when all the observations of the manuscript be solved.